# Obstetrics care in Indonesia: Determinants of maternal mortality and stillbirth rates

**Supriyatiningsih Wenang**[1]*, **Ova Emilia**[2], **Alfaina Wahyuni**[1], **Andi Afdal**[3], **Joerg Haier**[1,4]

**1** Faculty of Medicine and Health Sciences, Department of Obstetrics and Gynecology, Universitas Muhammadiyah Yogyakarta, Kasihan, Indonesia, **2** Faculty of Medicine, Department of Obstetrics and Gynecology, Public Health and Nursing, Universitas Gadjah Mada, Yogyakarta, Indonesia, **3** BPJS (Social Insurance Administration Organization), Jacarta, Indonesia, **4** Comprehensive Cancer Center Hannover, Hannover Medical School, Hannover, Germany

* supriyatiningsih.dr@umy.ac.id

## Abstract

### Problem

The Indonesian Healthcare Program starting in 2014 enabled access to healthcare delivery for large population groups. Guidance of usage, infrastructure and healthcare process development were the most challenging tasks during the implementation period. Due to the high social impact obstetric care and related quality assurance require evidence-based developmental strategies. This study aims for analysis of outcome and maternal health care utilization, as well as differences related to demographic and economic subgroups.

### Methods

For univariate group comparison ANOVA method was applied and combined with Scheffé procedure and Bonferoni correction for post-hoc tests. Meanwhile, multivariate approaches through regression analysis based on insurance reimbursement data antenatal, perinatal and postnatal care were performed at the province level. Maternal mortality (MMR) and still-birth rates were used for outcome. Demographic characteristics, availability of obstetricians (SPOG), midwifes and healthcare infrastructure were included for their determinants.

### Results

Specialized hospital facilities (type A/B) for advanced care covered a large part of uncomplicated cases (~35%). Differences between insurance membership groups (poor, non-poor) were not seen. Availability of human resources (SPOG, midwifes) ($R^2 = 0.728$; $p<0.001$) and rural setting ($R^2 = 0.288$; $p = 0.001$) are correlated with reduced insufficient referral. Their presence within provinces was related to lower occurrence of complicated cases ($R^2 = 0.294$; $p = 0.001$). However, higher SPOG rates within provinces were also related to high C-section rates ($p<0.001$). MMR and stillbirth rates can be predicted by availability of human resources and C-section rates explaining 49.0% of variance.

**Data Availability Statement:** Data from the national insurance BPJS are accessible via https://data.bpjs-kesehatan.go.id/bpjs-portal/action/landingPage.cbi Various restrictions regarding data

structure apply due to the legal framework of the insurance. Special requests can be addressed directly to the BPJS via the contact data on their website. Others would be able to access these data in the same manner as the authors and the authors did not have any special access privileges that others would not have.

**Funding:** The author(s) received no specific funding for this work.

**Competing interests:** The authors have declared that no competing interests exist.

**Abbreviations:** ANC, antenatal care; FKTP, Fasilitas Kesehatan Tingkat Pertama: First Level Health Facility; FKTRL, Fasilitas Kesehatan Tingkat Lanjut: Advanced Health Facility; INA-CPG, Indonesian diagnosis-related reimbursement system for inpatient treatment; JKN, Indonesian Healthcare Program; MDG's, Millennium Development Goals; MMR, Maternal mortality rates; PNC, postnatal care; SDG's, Sustainable Development Goals; SpOG, Spesialis Obstetri dan Ginekologi: Obstetrics and Gynecology Specialist; UHC, Universal health coverage.

## Conclusions

Improvement of perinatal outcome should focus on sufficient referral processes, availability of SPOG in provinces dominated by rural/remote demography and avoidance of overtreatment by high C-section rates. It is very important to regulate the education of obstetricians and gynecologists in Indonesia as well as distribution arrangements regarding to solve the problems with pregnancy complications in remote and rural areas.

## Introduction

While the novel Indonesian Healthcare Program (JKN) provides the background for achieving the universal health coverage (UHC) since 2014, this country continues to face significant challenges in meeting the sustainable development goals (SDGs) 4 and 5, particularly in lowering neonatal and maternal mortality rates (MMR) through preventive and curative efforts. In 2015 maternal mortality counted for 305 mother's death for every 100,000 population [1] Meanwhile in the 2020 population census, Indonesia's MMR has fallen to 189 maternal deaths per 100,000 livebirths [2], which is still high compared to other middle income countries in South-East Asia [3,4]. A high MMR is frequently associated to disparities of maternal health care access [5,6]. Indonesia faces challenges with access to healthcare for all parts of the community and inadequate healthcare workforces, particularly in rural areas [7,8]. The national physician-to-population ratio of 1:2294 and the availability of only approx. 5,000 gynecologists/obstetricians (SPOG) countrywide are associated with a MMR of 177/100,000 live births [9]. In 2020, the WHO UHC Index achieved via JKN was 60%, with high variability of the coverage values throughout the country [10,11]. Forseveral provinces their remote and isolated location causes low motivation for healthcare workers deciding to practice there, and many factors influence their recruitment in rural areas [10,12,13]. These regions face several barriers, such as infrastructure, difficulties to communicate, less quality of children's education, and low incomes [14]. Such scarcity and maldistribution of healthcare workers is enduring a significant issue worldwide due to unequal healthcare accessibility [7,12]. Furthermore, low- income regions are vulnerable to the concerns of the poor regarding their difficulties in accessing health services [4,15,16].

Many of the neonatal and maternal deaths could be avoided with antenatal care (ANC) interventions, and treatment of birth complications [17]. However, poor women population in remote and rural areas more often experience financial problems and limited access to health facilities resulting in low coverage of obstetric care, among others [6,13]. In the last few decades, Indonesia's policy to reduce maternal mortality since the implementation of the Millennium Development Goals MDG's resulted in a decrease in MMR and neonatal mortality rates. This decrease over the last few decades is most likely due to improving socio-economic conditions as well as government innovations related to targeting maternal health. Indonesia expanded the coverage of social health insurance, family planning, national midwifery program, introduced "birth waiting homes" to oversee deliveries in remote rural areas, and increased funding for puskesmas (Indonesia's primary care entities) [18,19]. A variety of issues have also been identified as opportunities to overcome observed gains [20]. Primarily, there should be better coordination in health program planning, management, and implementation [21,22].

The implementation of the JKN program by Indonesia can be considered as one of the most ambitious national healthcare schemes [18,22]. According to the national insurance

BPJS, enrollment rates increased from 48% in January 2014 to 85% by the end of 2019 covering 255 million citizensand approx. 60% of participants had to be classified as subsidized or poor and near-poor people.[24] Within this program ANC, labor and postnatal care (PNC), as well as referral care to hospitals are provided by JKN services [23–25]. JKN implementation in obstetrics healthcare services is still insufficient with unequal accessibility of services by Puskesmas or referral to advanced health care facilities (FKTRL). Since maternal mortality is likely associated to disparities of obstetric care access the development of national development strategies requires thorough nationwide analysis. Previous studies assessing Indonesia's obstetric care were still descriptive, fragmented, based on district-level population and lack of using nationally- representative data [26–28].

In this study we investigated determinants of the two main outcome aspects of pregnancy, MMR and stillbirth rates, related to demographic characteristics and availability of related healthcare structures in Indonesia. In addition, healthcare delivery processes were included as potential influencing variables. We used dataset sample from BPJS implementation period 2014–2018. Reimbursement data differentiated prenatal, perinatal and postnatal care with distinguished ICD10-based classification between uncomplicated and complicated cases.

## Methods

### Insurance data

Data about demographic, economy and healthcare infrastructure are available at public resources by the national statistical agency.[29] Reimbursement data (year 2018) were provided by BPJS as previously described.[8] Briefly, data were randomly selected from a family membership-based insurance registration. These data were provided as random selection from a family membership- based JKN registration (~73.4 member families). For each sample data an individual sampling weight was calculated and used in SPSS calculations (detailed description of the data extraction is provided as S1 File). Individual sampling weights were applied for statistical calculations. For selection of obstetrics care reimbursement diagnoses were limited to ICD10-codes O10-O48, O61- O99 and Z32-Z39. These ICD codes were further grouped into antenatal, perinatal and postnatal care. This resulted in 6 different ICD10-related treatment groups: Prenatal complications, Complicated delivery/perinatal care, Uncomplicated prenatal care, Complicated postnatal care, Uncomplicated delivery, Uncomplicated postnatal care. (Attribution of ICDs in Suppl. 2).

Definition of poor people was following the national guidelines in Indonesia and acknowledged according to their insurance schemes by PBI (poor) and non-PBI (non poor) members. The national classification system also provided an annotation of rural/remote regions at the provincial level (nationally referred to as 3T-regions, applicable in 27 of the 34 provinces). Previously, we reported clustering of Indonesian provinces according to their demographic characteristics into 3T poor, 3T non-poor and Non-3T provinces which was used for their comparison regarding demographic characteristics. [30]

Obstetrics care was differentiated into primary outpatient healthcare mainly provided by puskesmas and inpatient hospital care. BPJS reimbursement is based on capitation fees per insured member for outpatient care (FKTP type of reimbursement), procedure/diagnosis-related fees for primary care (non-capitation) and reimbursements for inpatient care according to respective effort (Indonesian diagnosis-related groups[INA-CPG], FKRTL reimbursement type). This reimbursement structure enabled differentiation between various scenarios of healthcare provision. All raw data on frequencies of obstetric care usage were aggregated at the provincial level and normalized per population in each province. For some evaluations,

esp. quadrant charts, obtainedusage rates were further standardized by dividing the mean of respective item (Item_X$_{MeanRel}$ = Item_X/mean $_{Item\_Xall}$).

Outcome data, such as MMR and stillbirth rates, were obtained from national statistics and used as dependent target variables. Since both variables reflect different aspects of the perinatal outcome they were analyzed separately in the multivariate approach. Furthermore, the frequencies of ICD- codes related to complications were selected and compared toall obstetric treatments within each province providing an indicator for complication rates. For investigation of caesarean sections the data were filtered for ICD10: O82; C-section rate was calculated by comparing the ICD10: O80.

## Statistical approach

**Analytical strategy.** First, we analyzed antenatal, perinatal and postnatal care for FKTP and FKRTL setting and differentiated between with and without complications according to the respective ICD codes. Treatment migration was determined as difference between the member's province of residence and province of treatment facility. Human resources for obstetrics included availability of board certified specialists in obstetrics / gynecology (SPOG) and midwifes per province as well as for 3T-regions.

Quadrant diagrams were constructed using two MeanRel-standardized items enabling visualization of relative data for the provinces. In these diagrams each dot represents one province. On both axis values of 1.0 represent the means of all provinces. Quadrants are attributes: HH (high values for X- and Y-axis), HL (high values for X- and low for Y-axis), LH (low X- and high Y-axis), LL (low values for both axis).

**Multivariate analysis.** MMR and stillbirth rate were used as dependent variables and all other data (demographic, insurance) were included as independent potential determinants. For univariate group comparison ANOVA method was applied and combined with Scheffé procedure and Bonferoni correction for post-hoc tests. Bivariate correlation was done to identify cross correlations that were excluded from further multivariate approaches to avoid collinearity (R>0.8).

To identify determinants for the outcome parameters stillbirth rate and MMR within the provinces linear regression was performed. All variables that fulfilled the univariate requirements were included in the beginning and stepwise excluded, if required. To ensure applicability of linear regression, linearity was obtained by transformative linearization of the variables if required. Then, a backward approach was applied to screen for items to be included into the final modelling. Using these items, a blockwise inclusion was done to obtain the final regression model in a more conservative approach. Fit of the regression was tested using ANOVA. Final acceptance of the regression model was obtained based on these formal criteria and premodelling. Correlation coefficients (non standardized and standardized) including their 95% confidence intervals are provided. Regressors were accepted once their contribution to the model was significant by Student's t-test (p<0.05). Heteroskedasticity, autocorrelation (Durbin/Watson) and multicollinearity (variance inflation factor) were excluded. $R^2$ is provided for explained variance and estimation for the fitness of the obtained regressions.

**Ethical approval.** Ethical approvals by the Universitas Muhammadiyah Yogyakarta (No. 202/EC-KEPK FKIK UMY/Vlll/2020) and the Indonesian National Healthcare Insurance BPJS (No. 5060/I.2/0419) were provided for the entire project. In this type of analysis informed consent was not applicable.

All statistical evaluations were done using IBM SPSS Statistics Version 26.

## Results

### Treatment groups

First, we compared the distribution of usage rates for the 6 treatment groups related to PBI/ Non-PBI membership and outpatient/inpatient settings. Under the outpatient primary care FKTP setting, 77.99% of the treatments were done for uncomplicated cases (PBI: 85.20%; Non-PBI: 70.77%) representing 6.02 Mio treatments (PBI: 3.01 Mio; Non-PBI: 3.01 Mio). This was related to antenatal care in 54.56% (PBI: 48.28%; Non-PBI: 60.64%), to perinatal care in 29.43% (PBI: 32.92%; Non-PBI: 25.95%), and to postnatal care in 16.01% (PBI: 18.80%; Non-PBI: 13.22%). (Fig 1A) For inpatient care (FKRTL), 52.96% of the 3.02 Mio treatments (PBI: 0.69 Mio; Non-PBI: 2.33 Mio) were done for antenatal care (PBI: 54.91%; Non-PBI: 46.17%), 37.09% in perinatal care (PBI: 35.63%;Non-PBI: 42.16%), and 9.95% in postnatal care (PBI: 9.46%; Non-PBI: 11.67%). In overall 55.72%, thiswas due to complicated diagnoses (PBI: 54.68%; Non-PBI: 59.32%). (Fig 1B) Significant differenceswere not observed comparing PBI and Non-PBI members (data not shown). Population based treatment rates were 2,343.8/ 100,000 for FKTP and 1,168.2/100,000 for FKRTL. However, there was extensive variations among the provinces (Range FKTP: 179.5–8,958.2; FKRTL: 328.7–4,900.6). (Fig 1C) Relative usage rates for both settings within the provinces correlated moderately ($R^2$ = 0.294). (S1A Fig in S3 File).

### Treatment migration

In the next step, usage of obstetrics care outside of the women's province of residence (treatment migration) was evaluated. This migration within the capitation outpatient scheme occurred in 7.05% (PBI: 1.56%; Non- PBI: 12.53%) of the treatments. For inpatient treatment, the referral rates from the various provinces varied between 4.32% and 22.95% of all treatments (complicated diagnoses in 45.17% of all referral cases). 11.36% of the patients were treated outside of their province of residence under inpatient conditions including 8.98% that were not referred from primary care settings. (Fig 1C).

### Usage of obstetric care facilities

Subsequently, the analysis focused on regional differences in the involvement of different types of healthcare providers in obstetrics care and the related outcome. If the different demographic province clusters were compared the usage of primary care structures for the various clinical settings was very similar. A significant difference was only found for uncomplicated postnatal care in 3T Non-Poor provinces (p = 0.018). (Fig 1E) The majority of inpatients was treated in Class C hospitals (46.91%) followed by Class B (25.95%) and Class D (14.16%). Specialized Mother & Child clinics and Class A hospitals were less frequently used (9.53% and 3.42%, resp.). Out of the 55.68% of complicated diagnoses 1.55% of the cases were treated in Class A hospitals, 15.41% in Class B, 25.89% in Class C, 7.73% in Class D, and 5.09% in Mother&Child clinics. (Fig 1F) Inpatient treatment for complicated cases was not associated with the relative rural population ($R^2$ = 0.034).

### Human resources for obstetric care

As second part of the resources as potential determinant of usage and outcome we investigated the regional distribution of staff availability. In Indonesia approx. 5,000 SPOG and 218,000 midwifes were available at the time of the usage analysis. Their distribution intensively differed between the provinces (SPOG: range 0–19.8/100.000population; midwifes: 36.2–213.0/ 100.000). In three provinces SPOG were not available at all. However, there was no correlation

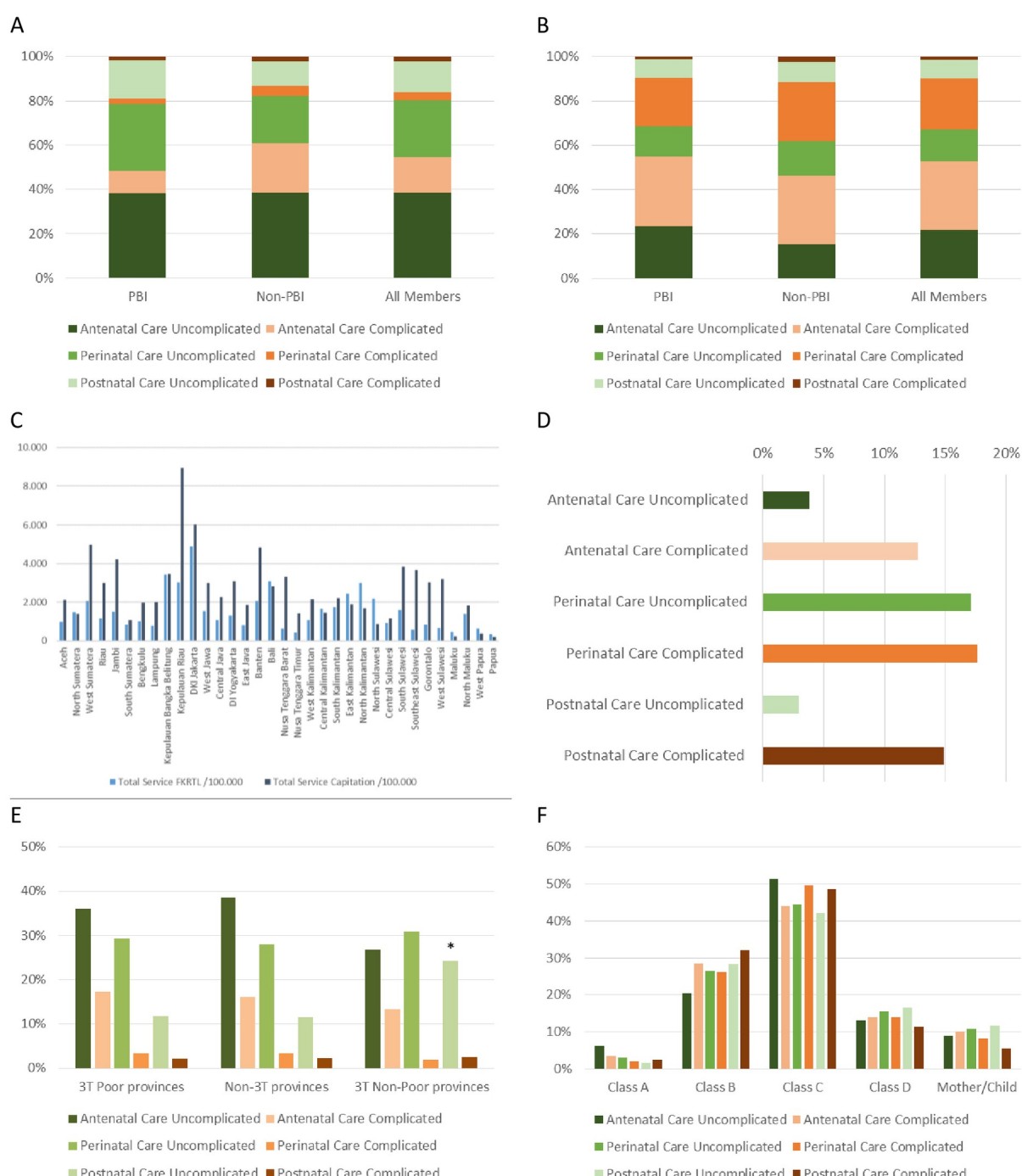

**Fig 1. Usage of obstetric care under.** A) primary (FKTP) and B) inpatient (FKRTL) setting; C) Population-based frequencies for treatment under FKTP and FKRTL setting; D) Migration of patients for advanced treatment (FKRTL) between provinces of residence and healthcare delivery; E) Usage of primary care (FKTP) in various demographic province clusters as percentage of care within the provinces. * p<0.05 compared to uncomplicated postnatal care in other province clusters; F) Relative involvement of hospital types in different clinical situations, percentage of all inpatients treated by different hospital types.

between the availability of both professional groups ($R^2 = 0.042$). Noteworthy, no provinces could be attributed to the HH- quadrant (high SPOG, high midwifes). (Fig 2A) Generally, especially in provinces with high rural population, the number of SPOG per population is very low. ($R^2 = 0.288$; p = 0.001; Fig 2B).

The availability of human resources within all provinces or in 3T regions was only in part related to the frequency of primary or inpatient care usage. For example, the population based frequency of complicated cases in PHC was not associated with the overall ($R^2 = 0.009$) or 3T ($R^2 = 0.045$) availability of midwifes (S2A and S2B Fig in S3 File) that was similar for inpatient treatment (S2C and S2D Fig in S3 File). However, for SPOG the inpatient frequency (similar for complicated and uncomplicated diagnoses) showed a strong correlation with their availability ($R^2 = 0.728$; p<0.001). However, if this frequency was corrected for the relative occurrence of complicated diagnoses within the overall obstetric care this association was not observed ($R^2 = 0.007$; Fig 2D). (S2E and S2F Fig in S3 File).

## Usage for clinical settings in rural and remote regions

As the last part of the potential determinants for obstetrics outcome we compared the usage rates with demographic frameworks (extent of rural population). The usage of primary care structures was highly different in the various provinces that was intensively related to the relative number of inhabitants living in rural areas. For treatment of complicated cases significant correlations were found between the FKTP usage and the relative numbers of rural population for overall ($R^2 = 0.294$; p = 0.001; Fig 3A), antenatal ($R^2 = 0.273$; p = 0.002)and perinatal ($R^2 = 0.218$; p = 0.006), but not for postnatal care ($R^2 = 0.004$; p = 0.089) (S2B-S2D Fig in S3 File). For inpatient care the population based frequencies of treatment for complicated diagnoses was similarly correlated with the extent of the rural population ($R^2 = 0.464$; p<0.001; Fig 3B). However,if these frequencies were related to the overall number of obstetric inpatient treatments this relationship was lost ($R^2 = 0.035$; Fig 3C).

## Clinical outcome

In order to target the main objectives of this investigation we evaluated the dependent outcome variables regarding their relationship to the potential determinants for better MMR and stillbirth rates. For the entire country, 4.77 Mio deliveries were recorded with a stillbirth rate of 4.00 per 1,000 (range for provinces 1.53–11.04/1,000). MMR for the provinces varied between 0.60 and 2.11/100,000 (mean 0.88/100,000) deliveries. These outcome parameters were not correlated with the population-standardized usage rates for obstetrics in primary or inpatient care (S3A-S3D Fig in S3 File). If these outcome parameters were related to human resources slight correlations were found. Stillbirth rates and MMR were significantly associated with SPOG ($R^2 = 0.124$; p = 0.001; Fig 4A; and$R^2 = 0.114$; p = 0.001; Fig 4B) and midwife availability ($R^2 = 0.159$; p = 0.001; Fig 4C; and $R^2 = 0.111$; p = 0.001; Fig 4D). However, for SPOG this was mainly due to the provinces Bali and Jakarta with very high numbers of SPOG in these provinces (quadrant HL). If these outlayers were not consideredthe correlation was lost (S4A and S4B Fig in S3 File). If this SPOG availability was provided in a population based manner this correlation was also not observed ($R^2 = 0.067$; S4C Fig in S3 File).

In the next step, we analyzed the influence of caesarean sections on the outcome parameters. The overall C-section rate was 48.7% (range: 18.5%–71.0%) of all single deliveries and its absolute number was strongly related to the number of available SPOG ($R^2 = 0.730$; p = 0.001; Fig 4E). This delivery type had a moderate significant relationship with the stillbirth rates ($R^2 = 0.227$; p = 0.001; Fig 4F) that was not found for the MMR ($R^2 = 0.096$; S4D Fig in S3 File). Similar to other delivery parameters the average numbers of C-sections per available SPOG

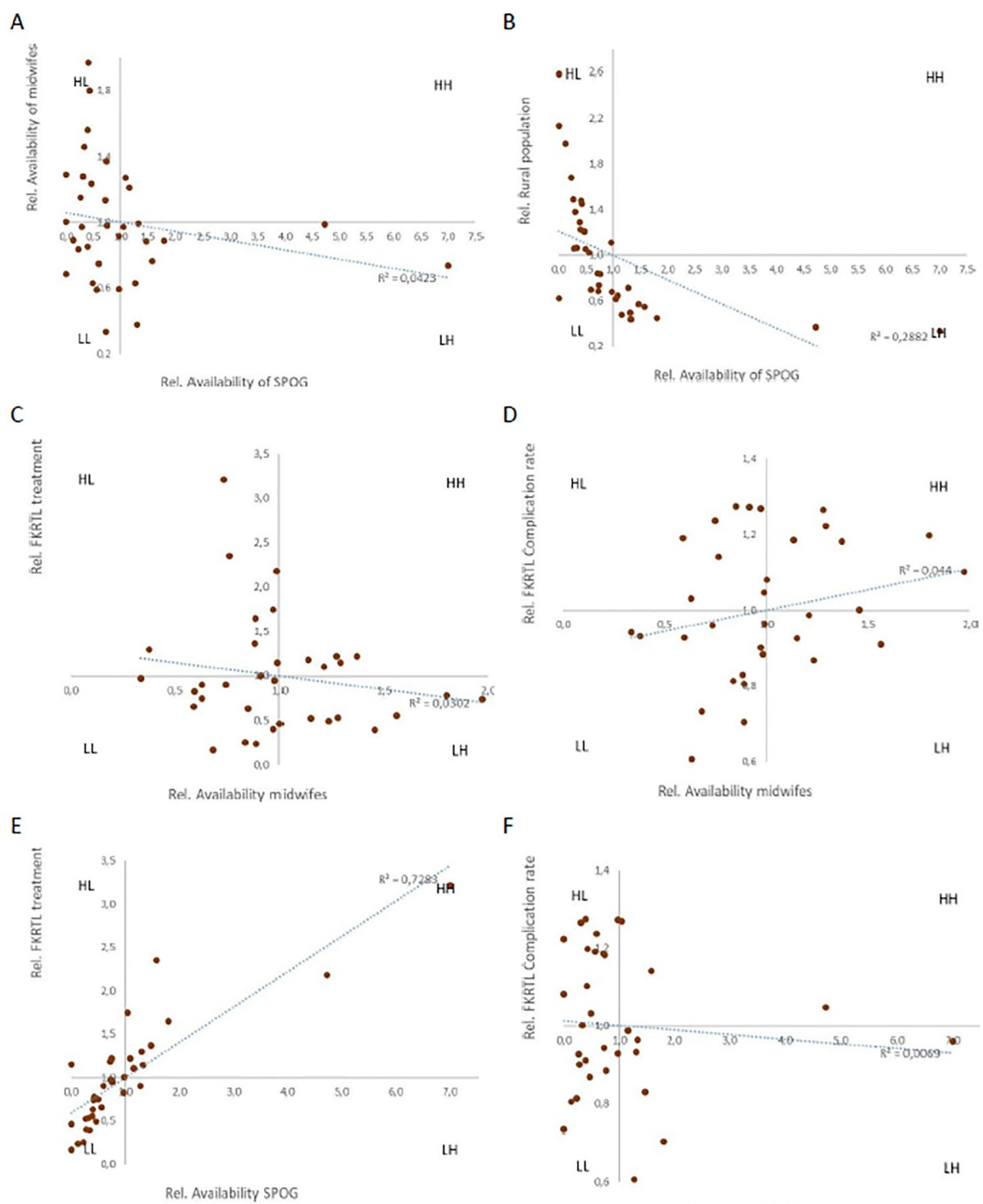

**Fig 2. Availability of resources.** Quadrant charts for A) availability of SPOG and midwifes; B) SPOG for rural population; Availability of midwifes: C) absolute frequency per 100.000 population and D) relative frequency of complicated compared to all diagnoses related; SPOG: E) absolute; F) relative frequency.

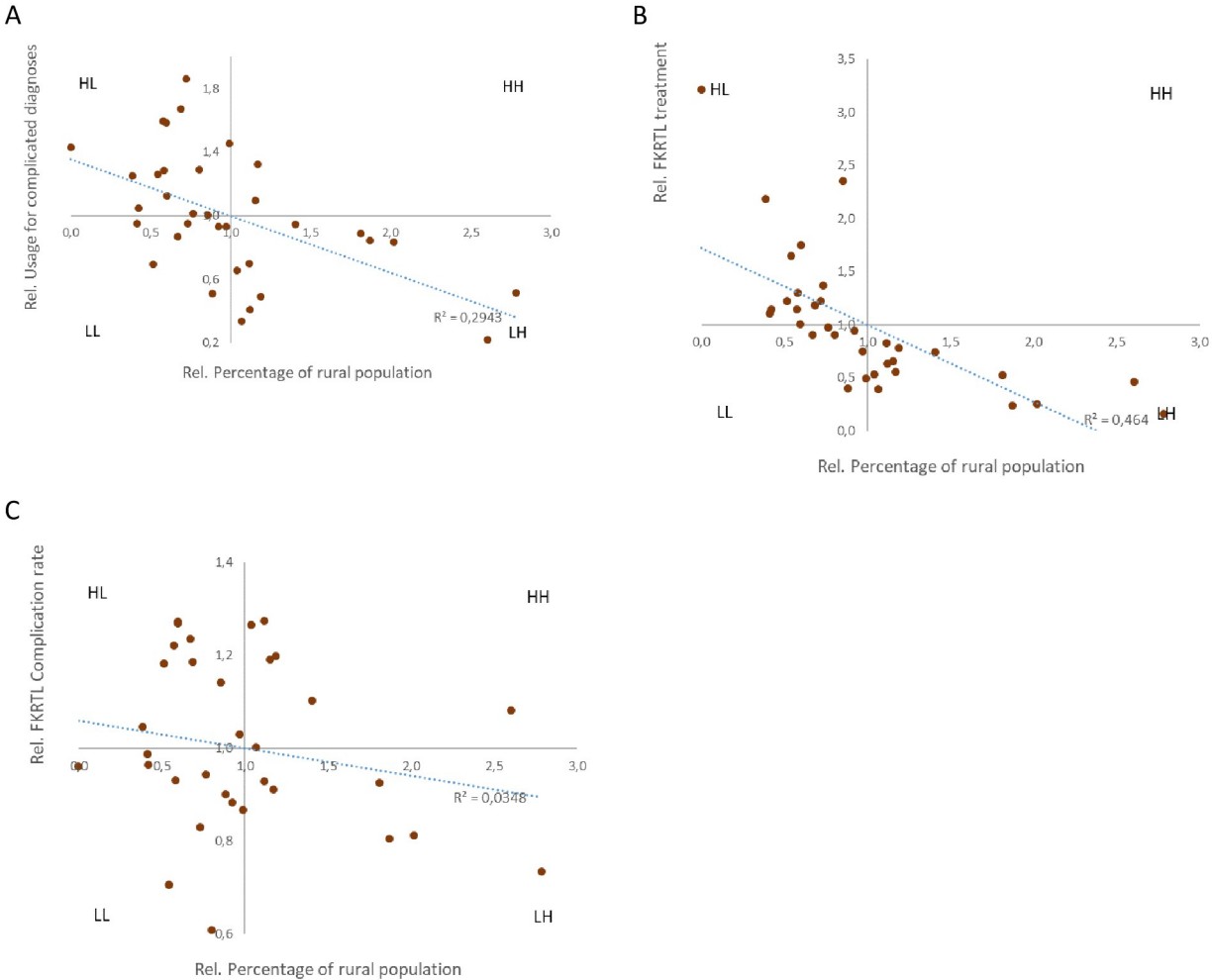

**Fig 3. Complicated diagnoses associated with relative numbers of rural population.** Quadrant charts for A) all FKTP treatment/rural population; (Differentiated data for antenatal, perinatal and postnatal care see S1B-S1D Fig in S3 File); B) absolute inpatient FKRTL treatment frequency per 100.000 population and C) relative frequency of complicated compared to all diagnoses for rural population.

showed a wide spectrum (mean 85.9; range 24.6–254.8). In two provinces, this average ratio was more than 200 C-sections/SPOG. Surprisingly, a moderate positive correlation was observed between this ratio and the MMR ($R^2 = 0.262$; $p = 0.001$; Fig 4F) that was not seen for the stillbirth rates ($R^2 = 0.075$; S4F Fig in S3 File).

## Multivariate analysis

Regression analysis was done to confirm factors that independently contribute to the high variability of the stillbirth and MMR between the provinces. All items showing significant correlations in the pairwise analysis were included in a backward approach. Interestingly, for both outcome parameters FKTP as well as FKRTL usage items did not show significant contributions to linear regression models (data not shown).

If the stillbirth rate was targeted as dependent outcome variable the resulting regression model (F test $p < 0.001$) contained three significant regressors (Delivery Single Caesarean Section (O82) Rate, HR All Regions Midwifes /100.000 and HR in 3T Regions Midwifes /100.000)

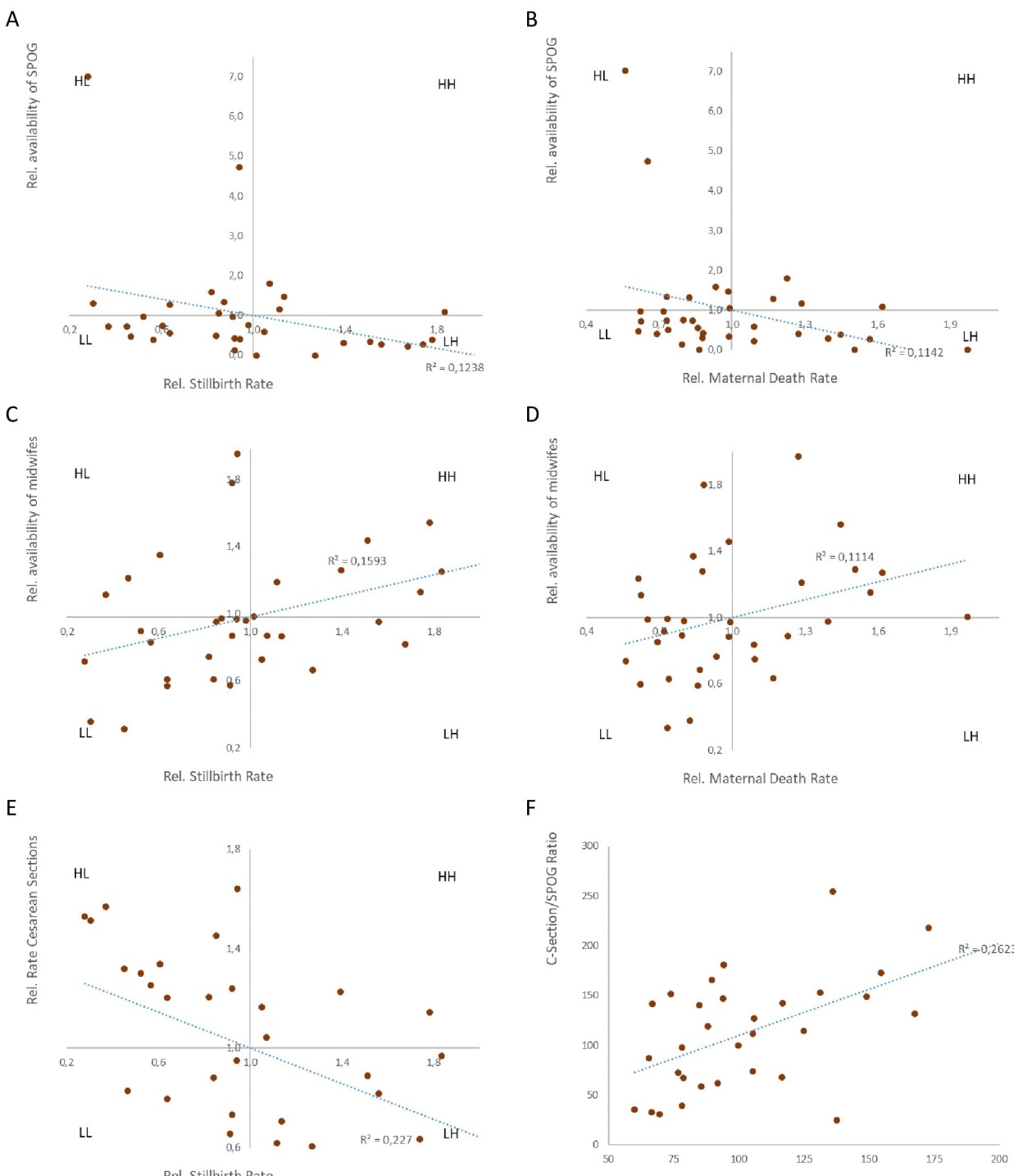

**Fig 4. Outcome related to availability.** Quadrant charts for gynecologists and midwifes (A; B) and advanced care (C; D) to stillbirth rates (A; C) and maternal mortality rates (B; D).

**Table 1. Linear regression model for A) stillbirth rates and B) MMR.**

| Items | Non standardized Coefficients | | Standardized Coefficients | | | 95% Confidence interval for B | |
|---|---|---|---|---|---|---|---|
| | Regression coefficient B | Std.-Error | Beta | T | Sig. | Lower limit | Upper limit |
| **A) Dependent variable: Stillbirth rate** | | | | | | | |
| (Constant value) | 3.631 | 1.469 | | 2.471 | 0.021 | 0.592 | 6.670 |
| Delivery Single Caesarean Section (O82) Rate | -0.063 | 0.023 | -0.373 | -2.703 | 0.013 | -0.112 | -0.015 |
| HR All Regions Midwifes /100,000 | 0.030 | 0.009 | 0.439 | 3.275 | 0.003 | 0.011 | 0.048 |
| HR in 3T Regions Midwifes /100,000 | 0.015 | 0.004 | 0.493 | 3.623 | 0.001 | 0.007 | 0.024 |
| **B) Dependent variable: MMR** | | | | | | | |
| (Constant value) | 1.067 | 0.138 | | 7.725 | 0.000 | 0.784 | 1.350 |
| Number of C-Sections per SPOG | 0.004 | 0.001 | 0.752 | 4.982 | 0.000 | 0.002 | 0.006 |
| Delivery Single Caesarean Section (O82) Rate | -0.011 | 0.003 | -0.534 | -3.540 | 0.001 | -0.018 | -0.005 |

and explained 59.9% of the observed variance. Its final regression model is shown in Table 1A demonstrating that the availability of midwifes was increasing this outcome whereas high C-section rates decreased the stillbirth rates.

Using maternal mortality as predictive target only two variables were identified as significant regressors (Number of C-Sections per SPOG, Delivery Single Caesarean Section (O82) Rate) explaining 49.0% of variance. High C-section rates again decreased maternal mortality, but if these C-sections were done by few SPOG (high C-section/SPOG rates) a negative impact on this outcome was identified. The standardized Beta-values showed that the impact of these regressors for both outcome variables is only moderate.

As shown above, two provinces (DI Jakarta, Bali) were specifically characterized by their availability of SPOG and the rates of C-section and overall deliveries. Therefore, all analyses were done excluding these potential outlayers, but multivariate results were similar. In addition, we differentiated inpatient usage rates for complicated diagnoses according to the hospital classes, but significant impact on regression functions were also not found. (data not shown) Moreover, the previously described clustering of the provinces (demographic and general healthcare resources) did not contribute to the explanation of the outcome variance between the provinces.

## Discussion

This analysis presents the results of obstetric services in JKN scheme. It identified various characteristic of the obstetric care usage throughout Indonesia. Although different hospital classes have dedicated tasks regarding specialized (class A&B) and non-specialized care (class C&D) there were very similar services rates without differentiation between complicated and uncomplicated diagnoses and almost half of all inpatient cases were treated in class C hospitals. Differences between related usage behavior between PBI and Non-PBI members were not seen, but treatment migration between provinces for inpatient obstetric care was almost exclusively done by Non-PBI members. This is in accordance with the current conditions in Indonesia where obstetric care usage in several large cities, such as Surabaya, Medan, Bali and Makasar, also shows that non-PBI patients prefer migration treatment. The urban-centered mobility occurs since non-PBI patients have more access to move from one service center to another [7]. In a previous survey in Indonesia in the pre- insurance period had similarly reported dependence from demographic characteristics [29–31]. In a Indonesian survey that was

conducted at the same time (Indonesian Basic Health Research Study: *Riskesdas*, 2018) confirmed that the optimal utilization of maternal health services is significantly influenced by determinants, such as age at pregnancy, mother's education, health insurance, parity, place of delivery, access to health facilities, urban-rural location and region [32].

For obstetric care low rates of SPOG and intermediate rates of midwifes are available, but very large differences occur between the provinces that are related to obstetric outcome. Inpatient treatment frequency was clearly determined by SPOG per population, but was not related to the occurrence of complicated diagnoses, stillbirth or MMR. The high variability of the stillbirth (~7-fold) and maternal mortality (3.5-fold) between the provinces was mainly determined by midwife availability and the frequency of C-sections. This is not surprising considering that the availability of human resources, especially the number of SpOG per population and midwife, plays an important role in influencing pregnancy outcomes. High C-section rates per population predicted better outcome, but the outcome was negatively associated if this was done by small numbers of SPOG (high rates of C- sections/SPOG). Especially, for maternal mortality these inverse effects of C-section rates and C- sections/SPOG had the highest impact in the regression model. The overall and 3T-availability of midwifes in the provinces was a predictor for high stillbirth rates, but further associations explainingthis phenomenon could not be verified in the regression analysis. The availability of competent health workers in Indonesia is essential to proper functioning of the health system [33,34]. The value of health workers in rural areas has been ever more highlighted by their impact on rural populations as public health requirement. Surprisingly, the differences in FKTP as well as in FKRTL usage rates between provinces with large rural or mainly urban population did not result in any impact on the average outcome variables.

The distribution of the poor population is becoming increasingly important for healthcare management and coverage. By dividing obstetrics usage data in Indonesia's rural areas into 4 quadrants, it can be seen that usage and outcome are predominated by two environmental settings: HL (High usage in low rural environment) and LH (Low usage in high rural environment) with significant relation to outcome (MMR, stillbirth rate). Therefore, healthcare care development in Indonesia needs critical attention towards obstetric services access, especially in terms of procuring human resources such as SPOG and midwives in rural areas [10,15].

The inappropriate dissemination between primary and advanced care usage for uncomplicated and complicated cases appears to be a key factor for improvement of effectiveness and efficiency of obstetric care in Indonesia. Services carried out in type A and B hospitals that can actually be carried out in type C and D hospitals have an impact on the large expenditure of expenses by the national health insurance. Class A hospitals have the highest costs when compared to other types and are actually reserved for severe cases [18,29]. Therefore, it seems to be necessary to increase the clarity of the referral rules and real-life implementation of related processes in the use of community-wide national health insurance. Successful implementation of suitable and accepted referral systems requires support from central government, national health organizations, physicians and patients who must play an active role in carrying out the country's health policy for improving health care quality [13,32].

In a country like Indonesia with highly challenging topography implementing telemedicine approaches for primary healthcare accessibility and referral advice appears to be of utmost importance [8].

Besides the lack of sufficient referral processes the overuse of advanced care for uncomplicated cases might be caused by limited acceptance of primary care and ease of access to advanced care in some regions. JKN member participation has enormously increased within a short period of time but equal dispersion of healthcare facilities in rural and remote regions is still hard to achievemaking it difficult for poor people living there to obtain these services

[22,35,36]. Though provision of expected quality of treatment throughout the country might be also challengingbut was not investigated in detail in this study [2,14]. Furthermore, the accessibility of obstetrics care is a constant constraint [20,25]. For example, a recent investigation in the province of Maluku, Indonesia, showed that services for maternal and child health in rural areas experienced various obstacles, such as unstable access to security, hindered by weather conditions, inadequate human resources, and inadequate hospital facilities [10]. The specific challenges that the country has to face in tis healthcare development were also found in our previous analyses. Although the differences can be seen in the various clinical fields, such as shown for primary care and cancer care, the overall requirement for trust building into the new healthcare delivery system for acceptance and development of sufficient human resources for availability and accessibility appears to remain crucial in Indonesia [8,31].

The investigation has some methodological shortcomings. BPJS insurance reimbursement data reflect usage of healthcare structures which are closely related, but not identical with incidences. The available data structure was dependent from the financial perspective of the insurance. Currently, quality of care is included in a very limited manner and only ICD10-based information regarding complication rate and outcome could be included in this investigation. Patient selection bias was addressed by a rigorous procedure that has been used in various previous analyses [8,31]. The implementation period of BPJS enrolment was characterized by differences between provinces and population groups but membership coverage of >70% throughout the country at time of data acquisition enables general conclusions for subsequent years with higher coverage rates [8]. For the statistical investigation we assumed linearity and limited the variables for regression analysis resulting in a potential risk to oversee factor interactions. However, the high number of variables in the initial steps and the comparison at the provincial level (N = 34) may have an even higher modelinguncertainty in a nonlinear approach. Therefore, the authors are convinced that assumption of linearity is sufficient for identification of relevant determinants of obstetric care.

In summary, obstetric services in Indonesia according to JKN scheme are dominant in outpatient primary care setting. Service distribution is similar in PBI and Non-PBI membership groups andbetween different province clusters. However, healthcare hospital facilities (type A/B) for advanced care covered a large part of uncomplicated cases that should have been carried out in type C and D hospitals. Availability of human resources (SPOG, midwifes) is strongly interlinked with reduced insufficient referral and their presence within provinces was related to lower occurrence of complicated cases. However, higher SPOG rates within provinces were also related to high C-section rates. Further improvement of perinatal outcome should focus on sufficient referral processes, availability of SPOG in provinces dominated by rural/remote demography and avoidance of overtreatment by high C-section rates. It is very important to regulate the education of obstetricians and gynecologists in Indonesia as well as distribution arrangements regarding to solve the problems with pregnancy complications in remote and rural areas. The major limitation of our investigation is usage of secondary reimbursement related insurance data as source of treatment characteristics. These data have a financial background and are categorized, at least in part, by non-clinical aspects. Furthermore, the analysis focused on the implementation period of the national insurance between 2014–2018 thus requiring further evaluation for subsequent developments. However, this type of data and analytical approaches enables the coverage of the entire country that would be impossible using different data sources. Apart from that, future investigations can be improved once a structured documentation system is rolled-out by combining primary and insurance data as well as by evaluation at the district level for more in-depth conclusions.

## Supporting information

**S1 File. Description of data extraction, stratification and weighing.**
(DOCX)

**S2 File. ICD10 codes used in capitation based reimburesement (outpatient setting).**
(PDF)

**S3 File. Quadrant charts for usage of PHC structures under capitation setting.**
(DOCX)

**S4 File. Treatment groups and treatment migration.**
(DOCX)

## Acknowledgments

The authors acknowledge the BPJS health insurance for the provision of data and methodological support.

## Author Contributions

**Conceptualization:** Supriyatiningsih Wenang, Joerg Haier.

**Data curation:** Supriyatiningsih Wenang, Andi Afdal, Joerg Haier.

**Formal analysis:** Supriyatiningsih Wenang, Joerg Haier.

**Investigation:** Supriyatiningsih Wenang, Ova Emilia, Andi Afdal, Joerg Haier.

**Methodology:** Supriyatiningsih Wenang, Joerg Haier.

**Project administration:** Andi Afdal, Joerg Haier.

**Supervision:** Supriyatiningsih Wenang, Ova Emilia, Andi Afdal, Joerg Haier.

**Validation:** Ova Emilia, Joerg Haier.

**Writing – original draft:** Supriyatiningsih Wenang, Alfaina Wahyuni, Joerg Haier.

**Writing – review & editing:** Alfaina Wahyuni.

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
