## [Decision Letter · Decision Letter 0]

20 Sep 2023

PONE-D-23-09137Obstetrics care in IndonesiaPLOS ONE

Dear Dr. Wenang,

Thank you for submitting your manuscript to PLOS ONE. After careful consideration, we feel that it has merit but does not fully meet PLOS ONE’s publication criteria as it currently stands. Therefore, we invite you to submit a revised version of the manuscript that addresses the points raised during the review process.

1. Kindly consider revising the Title of your manuscript to better reflect what was included in this study and reported in the manuscript.2. Kindly revise your abstract to include research objectives, statistical methods, including sampling methodology, results and conclusions derived from the study.3. Kindly revise your introduction to provide a more in-depth background to the study including research objectives and the aims of this particular study. Also, fully explain unclear abbreviations the first time used, before using abbreviations in the rest of the manuscript. If necessary you may include a table of abbreviations at the end of the manuscript.4. In the methods section kindly revise and better explain the statistical approach and parameters used in the study, including sampling methodology.5. In the Results section kindly revise to better explain the findings from the study and its interpretation.6. Kindly revise your discussion in the light of the findings from your own study in comparison to other recent and similar studies in Indonesia, and elsewhere in the literature.7. Insert a Statement Limitations regarding the findings from this study and their potential for generalization and areas for future study.8. The authors may consider using an English language editor to address all the grammatical and typographical errors in the manuscript.8. Kindly address all other comments and observations as outlined by the peer-reviewers.

We look forward to receiving your revised manuscript.

Kind regards,

Sylvester Chidi Chima, M.D., L.L.M, LLD.

Academic Editor

PLOS ONE

Journal Requirements:

Reviewers' comments:

Reviewer's Responses to Questions

**Comments to the Author**

1. Is the manuscript technically sound, and do the data support the conclusions?

Reviewer #1: Partly

Reviewer #2: Partly

2. Has the statistical analysis been performed appropriately and rigorously? 

Reviewer #1: No

Reviewer #2: Yes

3. Have the authors made all data underlying the findings in their manuscript fully available?

Reviewer #1: No

Reviewer #2: No

4. Is the manuscript presented in an intelligible fashion and written in standard English?

Reviewer #1: No

Reviewer #2: Yes

5. Review Comments to the Author

Reviewer #1: In Methods section, the authors should:

1. edit one reference citation (Methods section, 2nd paragraph);

2. mention the classic assumptions need to be met for linear regression and/or ANOVA. One of them is diagnostic test of the residuals and mention the criteria for choosing the best model;

3. use of the coefficient of correlation instead of the coefficient of determination in explaining bivariate correlation.

4. mention more specific the independent and dependent variables in their research;

5. mention the reason of the choice of those variables in point 4.

6. explain why the independent variables for each outcome variable in this research are different.

7. further discuss the connection between the results of this study with the previous studies.

8. review the writing style, especially when using fullstop (.) before refering a certain figure or table (example: Results section, 1st paragraph, 2nd sentence).

9. write one paragraph consists of minimum three sentences.

Reviewer #2: • This study presents an overview of obstetric care in Indonesia in terms of the use of health insurance organized by the Indonesian government, namely jaminan kesehatan nasional (JKN). The authors want to explain obstetric services in Indonesia based on this insurance scheme (JKN) in providing protection for maternal health services, especially in increasing access to obstetric services for low-income families with various challenges related to coverage and services.

• Although it was not explicitly explained regarding the novelty of this study, the authors explained that this study was slightly different from the previous studies by using national-scale dataset sample, namely from BPJS implementation for the 2014-2018 period to analyze the outcomes and utilization of maternal health services, as well as differences related to demographic and economic subgroup.

• However, there are still several principles of scientific writing that are not consistent and still receive less attention from a feasibility perspective. Important things that need to be considered and consistent start from identifying problems, developing research methods, analyzing and interpreting research results to drawing conclusions and provide recommendations to readers.

• Furthermore, to improve the quality of this article, it is recommended that the authors consult further with a biostatistician or epidemiologist. The authors should also pay more attention to the many sentences that are still not written correctly.

Comments to the authors:

Title:

According to the title of this article, namely “Obstetrics care in Indonesia”, the information that is expected to be available is various aspects related to obstetric care, including several key aspects of obstetrics care in Indonesia, for example prenatal care, birth facilities, emergency obstetric care, postpartum care, skilled birth attendants, and maternal health insurance. However, in this study, only some of the problems related to obstetrics care were actually reported regarding maternal health insurance. As a suggestion, the authors should reconsider the title of this study to be more specific and explain exactly what is being studied.

Abstract

In the abstract section, although it is limited by the number of words but it is important to explain clearly about the problem and research objectives to be answered. This is not been explicitly stated in this section. Apart from that, in the methods section, there is no explanation about the study design, the samples participating in this research and how to determine them, as well as the instruments or data collections tools even though the authors use secondary data. Furthermore, the author has not explained in the conclusion section whether the research has practical implications that can be applied in a real context.

After reading the entire article, perhaps the important findings of this study are: “The high variability of stillbirth rates (~7 times) and maternal mortality rates (3.5 times) between provinces is mainly determined by the availability of midwives and the frequency of caesarean sections. A high caesarean section rate per population predicts better outcomes, but outcomes are negatively associated if these are performed with a small number of SPOGs (high caesarean section/SPOG rate).” This is actually clear in the body of the results and discussion section, but not clearly visible in the abstract section. It should be noted that the abstract should provide a concise but clear description so that the reader can understand the contents of the article without having to read the whole article.

Introduction

In the introduction section, the authors have presented various background on the research topic. However, the authors have not clearly identified gaps in the research by reviewing existing related literature and comparing them with previous research both conducted in Indonesia and abroad on relevant topics, and are still lacking in explaining why this research needs to be carried out to fill the gap. The authors also still uses several references that are not relevant and up to date.

In addition, the research objectives were not clearly conveyed, what was to be achieved through this research and why yours research needs to be carried out.

The authors stated, “Currently, maternal mortality counts for 305 mother’s death for every 100,000 population,” where this data has been taken from two references published in 2018. According to United Nations Population Fund (UNFPA), Indonesia’s MMR at 305 is the MMR in the 2015 according to Inter-Censal population Survey data (https://indonesia.unfpa.org/en/topics/maternal-health-6). Meanwhile in the 2020 population census, Indonesia’s MMR has fallen to 189 maternal deaths per 100,000 livebirths, although it is still substantially higher compared to other Southeast Asian countries (https://indonesia.unfpa.org/en/news/strengthening-data-reduce-maternal-deaths-indonesia). Even inconsistently the authors took reference number 9 which reported Indonesis’s MMR at 177/100,000 live births.

In the Introduction and other sections, the authors have written abbreviations that are not international in nature which are actually taken from Indonesian abbreviations, for example JKN, SpOG, T3, FKTP and FKRTL. The authors are advised to explain in more detail so that readers, especially non Indonesians will more easily understand these terms.

Methods

The authors have explained that the data used in this study was retrospective data published by the national statistical agency and the insurance dataset for 2018 from BPJS. However, in this methods section, the authors have directly explained about the data without providing an explanation of the research approach used and how this approach will help achieve the research objectives. The authors also did not provide information about the study design and why it was suitable for answering the research questions. There is no further explanation regarding the data collection procedures carried out by national statistical agency and BPJS, and how the validity and reliability of these data.

The authors stated, “Briefly, data were randomly selected from a family membership-based insurance registration. Individual sampling weights were applied for statistical calculations.” However, it is not been clearly explain in a systematic structure, what data are used, what are the independent and dependent variables analysed, including how many provinces dan how many participants are analysed per outcome.

Statistical approach

In this section, authors stated that “Quadrant diagrams were constructed using two MeanReal-standardized items enabling visualization of relative data for the provinces”. In this context, MeanRel-standardized likely refers to a process of standardizing the means of the constructs relative to their standard deviation. These diagrams provide a way to access and compare the relative position of different constructs based on their average values and variability. However, in this section the author do not explain the basis for using the MeanRel-standardized, including what variables and outcomes are assessed. Looking at the figures provided in the attachment, it can be seen that are 3 outcomes assessed: 1) migration treatment; 2) Usage of obstetric care facilities; and 3) “untitled”, although this is inconsistently explained.

Results

The authors have presented and interpreted the research results based on the analysis carried out by describing and also displaying them in the form of graphs and figures to make it easier for reader to see the findings that emerge. However, the authors have not presented the results clearly and systematically. This may be because the authors did not clearly define the research objectives and research questions they wanted to answer.

For example, in the explanation on the treatment group section, it is quite difficult to understand what exactly the authors want to present, and what are the main findings. After looking at Figure 1A (although this figure, and other figures are not given a clear title), perhaps the important finding that should be interpreted is that complicated antenatal care services are used more by non-PBI and uncomplicated perinatal care by PBI participants.

Likewise, the interpretation of other results (treatment migration, usage of obstetric care facilities, human resources for obstetric care, usage for clinical settings in rural and remote regions, and clinical outcomes) they be able to interpreted clearly, objectively, and systematically in identifying the main findings that emerged from the data analysis.

In addition, in the results section the authors interpret the results by narrating the results and directing them to certain figures which have been written as figures 1 to 4. However, the figures shown in the appendix did not use the numbering 1 to 4 but were in the form letters A to F for all existing figures. This has the potential to cause difficulties for the reader.

Discussion

In the discussion section, the authors have summarized, interpreted, and argued the findings. Indonesia has made significant progress in improving maternal and child health over the years, but challenges still remain, particularly in remote areas and among marginalized communities. However, the authors still lack discussion in depth regarding the current research context by comparing it with previous studies and existing evidence. Identify similarities, differences, or contradictions between current findings and previous research. Explain how your findings make a new contribution or fill a gap in existing knowledge that support the arguments given.

The authors also need to discuss the implications of these findings in a broader context. Explain how the findings can contribute to existing theory, practice or policy. Identify the potential impact of your findings and explain why they are important.

It is worth considering that despite progress, Indonesia faces several challenges in obstetric care including disparities in access to services, inadequate infrastructure and equipment, shortages of skilled healthcare providers, and cultural practices that may hinder women from seeking professional care.

In fact, efforts are being made by the Indonesian government, as well as in collaboration with non-governmental organizations, and international partners to address these challenges and improve obstetric services throughout the country. So far, initiatives undertaken include training to improve the competencies and placing more specialist and midwives, improving health facilities, promoting community-based health programs, and increasing awareness about the importance of skilled birth assistance and prenatal care. This might be considered as a suggestion from the results of this study, compared to providing suggestions for regulating obstetrics and gynecology specialist education in Indonesia.

6. PLOS authors have the option to publish the peer review history of their article (what does this mean?). If published, this will include your full peer review and any attached files.

Reviewer #1: No

Reviewer #2: **Yes: **Windy Mariane Virenia Wariki

---

## [Author Response · Author response to Decision Letter 0]

14 Mar 2024

Response to reviewer

Editor

1. Kindly consider revising the Title of your manuscript to better reflect what was included in this study and reported in the manuscript.

2. Kindly revise your abstract to include research objectives, statistical methods, including sampling methodology, results and conclusions derived from the study.

3. Kindly revise your introduction to provide a more in-depth background to the study including research objectives and the aims of this particular study. Also, fully explain unclear abbreviations the first time used, before using abbreviations in the rest of the manuscript. If necessary you may include a table of abbreviations at the end of the manuscript.

4. In the methods section kindly revise and better explain the statistical approach and parameters used in the study, including sampling methodology.

5. In the Results section kindly revise to better explain the findings from the study and its interpretation.

6. Kindly revise your discussion in the light of the findings from your own study in comparison to other recent and similar studies in Indonesia, and elsewhere in the literature.

7. Insert a Statement Limitations regarding the findings from this study and their potential for generalization and areas for future study.

8. The authors may consider using an English language editor to address all the grammatical and typographical errors in the manuscript.

8. Kindly address all other comments and observations as outlined by the peer-reviewers.

Answer 

Thank you for the excellent revisions to this manuscript.

1. The title has been corrected to Obstetrics care in Indonesia: Determinants of maternal mortality and stillbirth rates

2. The abstract has been revised according to input

3. The introduction has been made complete with research objectives and added abbreviations at the end

4. The method has been revised by adding a statistical approach and parameters used in the study, including sampling methodology.

5. The results section has been given a more detailed interpretation

6. The discussion is equipped with references from several countries, especially Indonesia

7. Study limitations and further research that can be developed from this study have been given

8. grammatical and typographical errors in the manuscript have been revised

Reviewer 1

1. edit one reference citation (Methods section, 2nd paragraph);

2. mention the classic assumptions need to be met for linear regression and/or ANOVA. One of them is diagnostic test of the residuals and mention the criteria for choosing the best model;

3. use of the coefficient of correlation instead of the coefficient of determination in explaining bivariate correlation.

4. mention more specific the independent and dependent variables in their research;

5. mention the reason of the choice of those variables in point 4.

6. explain why the independent variables for each outcome variable in this research are different.

7. further discuss the connection between the results of this study with the previous studies.

8. review the writing style, especially when using fullstop (.) before refering a certain figure or table (example: Results section, 1st paragraph, 2nd sentence).

9. write one paragraph consists of minimum three sentences.

Answer 

Thank you for the excellent revisions to this manuscript

1. Citations have been edited

2. Modelling criteria were more extensively described in the methods section.

3. Correlation coefficients (non-standardized and standardized) including their confidence intervals are provided in table 1

4. Is added

5. Is added

6. In the beginning we used the same set of variables for both analyses, but during the stepwise procedures the obtained relevant variables were slightly different for the two outcome targets

7. We extended this topic in the discussion

8. Is modified

Reviewer 2

This study presents an overview of obstetric care in Indonesia in terms of the use of health insurance organized by the Indonesian government, namely jaminan kesehatan nasional (JKN). The authors want to explain obstetric services in Indonesia based on this insurance scheme (JKN) in providing protection for maternal health services, especially in increasing access to obstetric services for low-income families with various challenges related to coverage and services.

• Although it was not explicitly explained regarding the novelty of this study, the authors explained that this study was slightly different from the previous studies by using national-scale dataset sample, namely from BPJS implementation for the 2014-2018 period to analyze the outcomes and utilization of maternal health services, as well as differences related to demographic and economic subgroup.

• However, there are still several principles of scientific writing that are not consistent and still receive less attention from a feasibility perspective. Important things that need to be considered and consistent start from identifying problems, developing research methods, analyzing and interpreting research results to drawing conclusions and provide recommendations to readers.

• Furthermore, to improve the quality of this article, it is recommended that the authors consult further with a biostatistician or epidemiologist. The authors should also pay more attention to the many sentences that are still not written correctly.

Authors 

Answer 

Thank you for the excellent revisions to this manuscript

1. The author has added information related to obstetric services in Indonesia based on this insurance scheme (JKN) in providing protection for maternal health services

2. The author has added novelty of this study

3. The article writing has been revised

4. The author has consulted with all team members to improve this article

---

## [Decision Letter · Decision Letter 1]

5 Apr 2024

Obstetrics care in Indonesia: Determinants of maternal mortality and stillbirth rates

PONE-D-23-09137R1

Dear Dr. Wenang,

We’re pleased to inform you that your manuscript has been judged scientifically suitable for publication and will be formally accepted for publication once it meets all outstanding technical requirements.

Kind regards,

Sylvester Chidi Chima, M.D., L.L.M, LLD

Academic Editor

PLOS ONE

Reviewers' comments:

Reviewer #1: All comments have been addressed

Reviewer #2: All comments have been addressed

2. Is the manuscript technically sound, and do the data support the conclusions?

Reviewer #1: Yes

Reviewer #2: Yes

3. Has the statistical analysis been performed appropriately and rigorously? 

Reviewer #1: Yes

Reviewer #2: Yes

4. Have the authors made all data underlying the findings in their manuscript fully available?

Reviewer #1: Yes

Reviewer #2: Yes

5. Is the manuscript presented in an intelligible fashion and written in standard English?

Reviewer #1: Yes

Reviewer #2: Yes

6. Review Comments to the Author

Reviewer #1: (No Response)

Reviewer #2: Thank you very much for revising the article. After carefully read, I am pleased to inform that I fully support the rvision and approve for publication.

7. PLOS authors have the option to publish the peer review history of their article (what does this mean?). If published, this will include your full peer review and any attached files.

Reviewer #1: No

Reviewer #2: **Yes: **Windy Mariane Virenia Wariki

---

## [Editor Report · Acceptance letter]

3 May 2024

PONE-D-23-09137R1 

PLOS ONE

Dear Dr. Wenang, 

I'm pleased to inform you that your manuscript has been deemed suitable for publication in PLOS ONE. Congratulations! Your manuscript is now being handed over to our production team.

Kind regards, 

on behalf of

Professor Sylvester Chidi Chima 

Academic Editor

PLOS ONE